# Resistance to Fusarium Head Blight, Kernel Damage, and Concentrations of *Fusarium* Mycotoxins in the Grain of Winter Wheat Lines

**Piotr Ochodzki [1]**, **Adriana Twardawska [2]**, **Halina Wiśniewska [2]** and **Tomasz Góral [1],***

[1] Department of Plant Pathology, Plant Breeding and Acclimatization Institute—National Research Institute, Radzików, 05-870 Błonie, Poland; p.ochodzki@ihar.edu.pl
[2] Gene Structure and Function Team, Institute of Plant Genetics, Polish Academy of Sciences, 34 Strzeszyńska str., 60-479 Poznań, Poland; atwa@igr.poznan.pl (A.T.); hwis@igr.poznan.pl (H.W.)
* Correspondence: t.goral@ihar.edu.pl; Tel.: +48-22-733-4636

**Abstract:** Fusarium head blight (FHB) can contaminate cereal grains with mycotoxins. Winter wheat can also become infected with FHB and is more resistant than durum wheat to head infection but less than other small-grain cereals. The aim of this study was to identify winter wheat lines that combine low levels of head infection and kernel damage with low levels of grain contamination with mycotoxins. Resistance of 27 winter wheat lines (four with resistance gene *Fhb1*) and cultivars to FHB was evaluated over a three-year (2017–2019) experiment established in two locations (Poznań and Radzików, Poland). At the anthesis stage, heads were inoculated with *Fusarium culmorum* isolates. The FHB index was scored, and the percentage of *Fusarium*-damaged kernels (FDKs) was assessed. The grain was analyzed for type B trichothecenes (deoxynivalenol and derivatives and nivalenol) and zearalenone content. The average FHB index of both locations was 12.9%. The proportion of FDK was 6.9% in weight and 8.5% in number. The average content of deoxynivalenol amounted to 3.543 mg/kg, and the average amount of nivalenol was 2.115 mg/kg. In total, we recorded 5.804 m/kg of type B trichothecenes. The zearalenone content in the grain was 0.214 mg/kg. Relationships between the FHB index, FDK, and mycotoxin contents were highly significant for wheat lines; however, these relationships were stronger for FDK than for FHB index. Breeding lines combining all types of FHB resistance were observed, five of which had resistance levels similar to those of wheat lines with the *Fhb1* gene.

**Keywords:** ergosterol; *Fusarium culmorum*; mycotoxins; resistance; trichothecenes; zearalenone; winter wheat





## 1. Introduction

Mycotoxins have been widely studied in small-grain cereals worldwide. As secondary metabolites, mycotoxins are produced by fungal strains, mostly by *Fusarium*, *Aspergillus*, and *Penicillium* species. In wheat, nivalenol, deoxynivalenol, and zearalenone are the most common mycotoxins. These mycotoxins are produced by the pathogenic fungi *Fusarium graminearum* and *F. culmorum* and occur not only in grains but also in wheat-based products, which has detrimental effects on human and animal health [1,2].

Fungal diseases of bread wheat are one of the main factors that can lead to a decrease in grain yield and grain quality. An important group of fungi with parasitic–saprotrophic lifestyles that cause several cereal diseases, including in wheat, are species from the genus *Fusarium*. These fungi are the most important pathogens of wheat [3–6]. In wheat, Fusarium head blight (FHB) has caused billions of dollars in losses for farmers and grain marketing and processing companies in the United States [6–8]. FHB was found to be the main cause of yield losses in wheat in China. It was the second leading cause in the United States and Canada, and the third in southern part of South America [9].

Most information on the FHB of cereals relates to commonly grown wheat [6,10,11]. FHB causes the greatest damage to bread wheat and durum wheat. The latter cereal is the most susceptible to this disease [12]. FHB is referred to as a disease complex because it is caused by several species of *Fusarium*. The increases in the share of maize and winter wheat in cultivation, combined with simplified crop rotation, changes in cultivation techniques (zero tillage), and climate change, have increased the risk of this disease in recent years [13–16]. Many years of research on winter wheat varieties in Poland showed a small variation in susceptibility to FHB and indicated a lack among Polish cultivar genotypes with a low quantity of *Fusarium* mycotoxins in the grain [17–20].

*Fusarium* infection leads to the development of FHB, which appears on cereal spikes, where the pathogen can infect the entire spike through one spikelet [21]. FHB is characterized by an altered spike morphology, such as the appearance of brownish stains with spikelet bleaching and drying of the kernels as symptoms of premature spike dieback [22]. Additionally, pink–red sporodochia can appear on the spike's glumes. These sporodochia are the source of fungi spores and can represent an infection threat for neighboring plants [23].

The FHB severity is characterized by different resistance types: resistance to initial infection (type I), resistance to *Fusarium* spread within the spike (type II), resistance to kernel infection (type III), tolerance against FHB and trichothecenes (type IV), and resistance to trichothecenes (type V) via degradation or detoxification (class 1) or by preventing accumulation (class 2) [10,24].

*Fusarium* species produce numerous toxins that cause acute or chronic toxic effects in humans and animals, depending on the type of toxin and the amount of food or feed consumed [25]. In wheat kernel, previous toxicological analysis revealed the presence of two predominant toxin types: type B trichothecenes, such as nivalenol (NIV) and deoxynivalenol (DON), and zearalenone (ZEN) [4,26,27].

DON and NIV have strong toxic effects and can cause skin irritation, vomiting, diarrhea, appetite weakness, hemorrhaging, neurological disorders, miscarriages, and even death [28–30]. As stated by the World Health Organization (WHO), DON is considered to be a teratogen, neurotoxin, and immunosuppressive agent. DON together with NIV can cause diarrhea [31], nausea, and food refusal in nonruminant animals [32]. The toxicity of ZEN can disturb swine reproductivity via hyper estrogenic syndrome, which causes enlargement of the genitalia and mammary glands, as well as livestock infertility [33–35].

In many countries and regions, including the European Union, limits on the amount of *Fusarium* toxins in food and feed have been adopted. The purpose of establishing these limits have been the control of and reduction in mycotoxin contents in the grain of wheat and other cereals. These limits are governed by the most recent Commission Regulation (EC) No. 1126/2007 of 28 September 2007, which amended Regulation (EC) No. 1881/2006 and fixed the maximum levels of certain contaminants in foodstuffs, including *Fusarium* toxins in maize and maize products.

The way to reduce the content of *Fusarium* toxins may be to reduce the risk of infestation of wheat crops by *Fusarium*. This can be achieved by using fungicides and agrotechnical methods, including ploughing, crop rotation (a reduction in the share of cereals and maize), and the growth of resistant cultivars [36,37].

The most effective method to counteract *Fusarium* infection is to breed genetically improved cultivars with resistance to the pathogen. Resistance to FHB is a quantitative trait. Many minor genes in the *Triticum aestivum* genome offer additive effects [38] if they are inherited polygenically [39] as a quantitative trait locus (QTL). The main QTL is the well-examined *Fhb1* gene located on the short arm of chromosome 3B, which increases type II resistance in wheat [40,41]. The presence of this gene-resistant allele can reduce FHB severity by about 20% [42]. Moreover, *Fhb1* relates to plant detoxification due to the conversion of DON to the less toxic DON-3-*O*-glucoside [43]. Among the many reported QTLs [44], some also increase plant resistance; these QTLs include *Fhb2* [45], *Fhb4* [46], *Fhb5* [47], and *Qfhs.ifa-5A* [48].

The majority of resistant germplasms originated in China, from an area where numerous FHB epidemics have occurred [49]. Cultivar 'Sumai-3' and its consecutive derivatives have been the most commonly used in breeding programs, leading to elevated plant resistance levels [50,51]. Despite the intensive screening for resistant germplasms, genotypes with full resistance to FHB have not yet been found [8]. Environmental conditions such as high temperature and humidity foster *Fusarium* infection, which is important in the context of climate change [52]. Environmental changes can influence *Fusarium* species distribution due to pathogenic species shifts and expansion to a greater area [53,54]. Moreover, some *F. graminearum* isolates can more intensely respond to elevated temperature and $CO_2$ concentrations due to synthesizing a higher amount of DON and ZEA toxins [35]. This factor represents a significant threat to the future breeding of wheat, making it important to seek new resistant cultivars.

The aim of this work was to find winter wheat genotypes resistant to FHB by assessing the degree of head infection and damage to kernels by *Fusarium culmorum*. The selected genotypes with increased FHB resistance were tested for ergosterol (ERG), quantity of mycelium content, and resistance to the accumulation of *Fusarium* toxins (i.e., DON and its derivatives, NIV, and ZEN) in the grain. These analyses were conducted to identify genotypes that combine elevated levels of resistance to FHB, kernel damage, and mycotoxin accumulation in the grain.

## 2. Materials and Methods

### 2.1. Plant Material

Plant materials included 27 winter wheat lines and cultivars:

- Winter wheat cultivars: Arina [55], Artist (Deutsche Saatveredelung AG, Lippstadt, Germany), Fregata (Hodowla Roślin Strzelce Ltd.—IHAR-PIB Group, Strzelce, Poland), Patras (Deutsche Saatveredelung AG, Lippstadt, Germany), RGT Kilimanjaro (RAGT 2n, Rodez, France). The Artist, Patras, and RGT Kilimanjaro were high yielding check cultivars in the pre-registration testing system of the Research Centre for Cultivar Testing—COBORU. Available online: https://coboru.gov.pl (accessed on 24 August 2021) [56];
- Polish breeding lines of wheat: AND 4023/14, AND 82/11/50, DL325/11/3, KBP 14 16, NAD 10079, NAD 13014, NAD 13017, NAD 13024, POB 0616, POB 170/04, POB 679/03, SMH 7983, SMH 8694, SMH 8816, STH 008, and STH 9059;
- Lines of wheat resistant to FHB carrying the *Fhb1* gene: UNG 136.6.1.1 [Fhb1+], S10 [Fhb1+], S30 [Fhb1+], and S32 [FHB1+] [57,58];
- Lines of wheat resistant to FHB without the *Fhb1* gene: 20828, A40-19-1-2 [57,59,60].

Wheat lines originated from Polish breeding companies (DANKO Hodowla Roślin Ltd., Choryń, Poland; Hodowla Roślin Smolice Ltd.—IHAR-PIB Group, Smolice, Poland; Hodowla Roślin Strzelce Ltd.—IHAR-PIB Group, Strzelce, Poland; Małopolska Hodowla Roślin Ltd., Kraków, Poland; Poznańska Hodowla Roślin Ltd., Tulce, Poland) and were selected from a large set of breeding lines based on how low or high the head infection was in two environments (data not shown) [61]. Breeding lines and resistant lines were from the collection of FHB-resistant wheat of the Department of Plant Pathology, PBAI-NRI Radzików. The most resistant lines will be deposited at the National Centre for Plant Genetic Resources: Polish Genebank. Available online: https://bankgenow.edu.pl (accessed on 24 August 2021)

### 2.2. Fungal Material for Inoculation

The fungal material for inoculation included a mixture of three isolates of *Fusarium culmorum* (W.G.Sacc.): KF 846 (a DON chemotype), KF 350 (an NIV chemotype) derived from the collection of the Institute of Plant Genetics at the Polish Academy of Sciences (Poznań, Poland), and ZFR 112 (a DON chemotype that produces zearalenone) derived from the collection of the Plant Breeding and Acclimatization Institute—NRI (Radzików, Poland) [62].

Isolates were incubated on autoclaved wheat kernels in glass flasks for about 1 week at 20 °C in darkness and then exposed to near-UV light under a 16:8 h (light:dark) photoperiod for 3 weeks at 15 °C. The mycelium-colonized grain was air-dried and stored in a refrigerator at 4 °C until usage. At the date of inoculation, the grain with *F. culmorum* spores was suspended in water for about 2 h and then filtered through cheesecloth to obtain a conidial suspension. The suspensions from each of the three isolates were adjusted to 500,000 spores per milliliter with the aid of a hemocytometer (BRAND GmbH + Co. KG., Wertheim, Germany). Equal volumes of suspensions from the three isolates were then mixed.

### 2.3. Description of the Field Experiment

A three-year field experiment (2017, 2018 and 2019) was established in two locations. First was the experimental field of the Institute of Plant Genetics Polish Academy of Sciences in Poznań (field located 30 km north-west from Poznań, Poland; 82 m above sea level; GPS coordinates 52°31′21.3″ N 16°41′19.1″ E). Second was the experimental field of Plant Breeding and Acclimatization Institute National Research Institute in Radzików (central Poland; 87 m above sea level; GPS coordinates 52°12′45,4″ N 20°37′59.2″ E).

Experiments were established as a randomized block design. Wheat lines were sown in 1 m$^2$ (Radzików) or 0.5 m$^2$ (Poznań) plots in four replicates/blocks. Sowing occurred from the last week of September to the first week of October. Conventional tillage was applied in both locations at the end of August.

Pre-crop in Radzików was oilseed rape. Fertilizers were used according to standard agricultural practices. In the autumn, 3 dt/ha of Polifoska 6 (NKP(S) 6-2-30-(7)) fertilizer (Grupa Azoty Zakłady Chemiczne "Police" S.A., Police, Poland) was applied (N—18 kg/ha, P—60 kg/ha, K—90 kg/ha). In the spring, ammonium nitrate fertilizer (Grupa Azoty Zakłady Azotowe „Puławy" S.A., Puławy, Poland) was applied in an amount providing 70 kg N/ha. Weeds and pests were controlled with herbicides and insecticides. After sowing, weeds were controlled with herbicide Maraton 375SC (BASF SE, Ludwigshafen, Germany) (isoproturon + pendimethalin) in a dose of 4 L/ha. In spring, weeds and rape self-seeders were controlled using herbicide Attribut 70SG (Bayer CropScience AG, Monheim, Germany) (propoxycarbazone-sodium) in a dose of 60 mg/ha. Cereal leaf beetle and aphids were controlled with Fastac Active 050ME (BASF SE, Ludwigshafen, Germany) (alpha-cypermethrin) in a dose of 250 mL/ha. No fungicides were applied.

Pre-crop in Poznań was oilseed rape (2017, 2018) or lacy phacelia (2019). In the autumn, 4 dt/ha of Polifoska 5 (NPK(MgS) 5-15-30-(2-7)) fertilizer (Grupa Azoty Zakłady Chemiczne "Police" S.A., Police, Poland) was applied (N—20 kg/ha, P—60 kg/ha, K—120 kg/ha). In the spring, ammonium nitrate fertilizer (Grupa Azoty Zakłady Azotowe „Puławy" S.A., Puławy, Poland) was applied in an amount providing 70 kg N/ha. After sowing, weeds were controlled with herbicide Legato 500SC (ADAMA Polska Sp. z o.o., Warszawa, Poland) (diflufenican) in a dose of 1.5 L/ha. No fungicides were applied

### 2.4. Inoculation Procedure

At full anthesis (65 BBCH scale), from the end of May to a 10-day period in June, wheat lines were inoculated by spraying the heads with a spore suspension [63]. Three blocks of plots were inoculated, and a fourth non-inoculated plot served as a control. Inoculation was repeated three days later. Two days after inoculation, micro-irrigation was applied to maintain high moisture levels [62,64].

About two weeks after inoculation (depending on FHB symptoms appearance) and one week later, disease progress was visually evaluated using the FHB index (FHBi):

$$\text{FHBi} = \frac{\% \text{ of head infection } \times \% \text{ of heads infected per plot}}{100} \tag{1}$$

At harvest (end of July–beginning of August), 20 randomly selected heads from each plot (one control and three inoculated plots) at each location were collected and threshed with a laboratory thresher.

The percentage of *Fusarium*-damaged kernels (FDK) was scored visually according to the methods described earlier [65,66]. The FDK weight relative to the weight of the whole sample was marked as FDKw, and the FDK number relative to the total sample size was marked as FDK#.

*2.5. Toxin Analysis*

Wheat grain samples from three inoculated plots were mixed and finely ground. The concentration of *Fusarium* toxins in wheat grain was then analyzed using gas chromatography. Type B trichothecenes (DON, 3-acetyldeoxynivalenol (3AcDON), 15-acetyldeoxynivalenol (15AcDON), and NIV) were detected.

Mycotoxins were extracted from 5 g of ground grains using 25 mL of an aqueous solution of acetonitrile (acetonitrile:water, 84:16, *v/v*). Samples were shaken on the laboratory shaker overnight, centrifuged (3000 rpm/min, 5 min), and 6 mL of the extract was purified with MycoSep® 227 Trich+ columns (Romer Labs Inc., Union, MO, USA). Internal standard solution (α-chloralose, 1 mL, 1 μg/mL in acetonitrile) was added to 4 mL of purified extract and the solution was evaporated to dryness in a stream of air. Mycotoxins were derivatized to the trimethylsilyl derivatives with 75 μL of derivatizing agent, Sylon BTZ (BSA + TMCS + TMSI, 3:2:3, Supelco), and heated for 30 min at 60 °C. After the dissolution of the sample in 1 mL of isooctane, the excess of the derivatizing agent was decomposed and removed with water. The organic layer was transferred to an autosampler vial, and 1 microliter of solution was injected on GC.

Content of the trichothecenes of group B in the grain (DON, 3AcDON, 15AcDON, NIV) was analyzed using the gas chromatography technique. Gas chromatograph SRI 8610C (SRI Instruments, Earl St. Torrance, CA, USA), equipped with a splitless injector, [63]Ni electron capture detector (ECD) (VICI Valco Instruments, Schenkon, Switzerland), HT300A autosampler (HTA S.R.L., Brescia, Italy), HG 2200 (CLAIND srl, Tremezzo, Italy) hydrogen generator, BGB-5MS 30 m × 0.25 mm × 0.25 μm column (BGB Analytik Vertrieb GmbH, Rheinfelden, Germany) and PeakSimple data processing program were used.

The carrier gas was hydrogen, adjusted to a pressure of 12 psi, with nitrogen as a make-up gas at 60 mL/min. Elution was carried out in the following temperature gradient: initial temperature was 170 °C, increased to 250 °C at 5 °C/min, and increased from 250 °C to 300 °C at 10 °C/min, followed by a holding time of 5 min, and decreased to 170 °C. Individual compounds were identified by comparing the retention times of these with the retention times of the pure standards of mycotoxins (Biopure). The injection port and detector operated at 250 °C and 300 °C, respectively. The concentration of mycotoxins was established based on the calibration curve, using α-chloralose (Sigma-Aldrich sp. z o.o., Poznań, Poland) as the internal standard.

Preliminary studies on the analytical method (data not shown) revealed the repeatability (RSD) of the method, 7.2% for DON and 10.6% for NIV. Experiments on wheat meal spiked with pure mycotoxins (1000 μg/kg of each) showed recoveries of DON—93% and NIV—72%. All results were corrected for recovery.

The content of ZEN was determined using a quantitative direct and competitive enzyme-linked immunosorbent assay (ELISA) AgraQuant® Zearalenone 25-1000 (LOD 20 ppb, LOQ 25 ppb) (Romer Labs GmbH, Tulln, Austria). A 5 g ground sample was placed in a conical 50 mL Falcon centrifuge tube; then, 25 mL of the solvent (methanol–water 70:30 *v/v*) was added. The sample was extracted for 1 h on a shaker and then centrifuged (1620 g, 5 min). The obtained extract was analyzed with the ELISA method according to the procedure described by Romer Labs. The content of ZEN was expressed as toxin weight (mg) per grain weight (kg).

Ergosterol was chromatographically analyzed with methanol using the HPLC technique on a silica column. Ergosterol (ERG) was chromatographically analyzed via high-

performance liquid chromatography (HPLC) on a silica column using methanol. A detailed evaluation of the method is given in the paper by Perkowski et al. [67]. Samples containing 100 mg of ground grains were placed into 17 mL culture tubes, suspended in 2 mL of methanol, treated with 0.5 mL of 2 M aqueous sodium hydroxide, and tightly sealed. The culture tubes were then placed within 250 mL plastic bottles, tightly sealed, and placed inside a microwave oven operating at 2450 MHz and 900 W maximum output. Samples were irradiated (370 W) for 20 s, and then after approximately 5 min, for an additional 20 s. After 15 min, the contents of the culture tubes were neutralized with 1 M aqueous hydrochloric acid, 2 mL MeOH was added, and extraction with pentane (3 × 4 mL) was carried out within the culture tubes. The combined pentane extracts were evaporated to dryness in a nitrogen stream. Before analysis, samples were dissolved in 1 mL of MeOH, they were filtered through 13 mm syringe filters with a 0.45 μm pore diameter (Fluoropore Membrane Filters, Millipore, Ireland), and 50 μL was injected on the HPLC column. Separation was performed on a reversed phase column Nova Pak C-18 (Waters, Milford, MA, USA), 150 × 3.9 mm, particle size 4 μm, and eluted with methanol/acetonitrile (90:10) at a flow rate of 0.6 mL/min. Ergosterol was detected with a Waters 486 Tunable Absorbance Detector (Milford, MA, USA) set at 282 nm. The presence of ergosterol (ERG) was confirmed by a comparison of retention times and by the co-injection of every tenth sample with an ergosterol standard.

### 2.6. Statistical Analysis

The statistical analysis was performed using XLSTAT© Life Science, Version 2021.2.1.1119 (Addinsoft, New York, NY, USA).

FHB and FDK ratings, reductions in yield components, concentrations of ERG, and toxin data were analyzed through analysis of variance (ANOVA) procedures using the XLSTAT procedure. Three-way (year × location × line) ANOVA was performed. Next, two-way ANOVAs were conducted. Year × line separately for two locations and location × line separately for three years. In all ANOVAs, the year effect was considered as random, and the location and line were considered as fixed. Normality of the data distribution was tested using a Shapiro–Wilk test (XLSTAT procedure: Normality test). All variables did not follow normal distribution and were transformed with Box–Cox (FHBi, FDKw, FDK#) or $\log_{10}$ (ERG, DON, 3ADON, 15AcDON, NIV, TCT B, ZEN) transformations. Mean differences were determined according to Fisher's LSD test at $\alpha = 0.05$.

The relationships between FHBi, FDK, ERG, and mycotoxin concentrations were investigated by Pearson correlation tests (XLSTAT procedure: correlation tests). Prior to analysis, variables (means for the 27 lines) that did not follow a normal distribution were $\log_{10}$-transformed to normalize residual distributions.

The multivariate data analysis method was applied to the data on FHB resistance (FHBi, FDK#, ERG, DON, NIV, ZEN). Principal component analysis (XLSTAT procedure: Principal Component Analysis PCA) was used to show how wheat lines were distributed with respect to the main variation described in the first two components and how variables influenced the construction of the two components. PCA results also revealed associations among variables measured by the angle between variable vectors.

## 3. Results

The average severity of FHB was FHBi = 12.9%; FHBi was greater in Radzików (15.2%) than in Poznań (9.4%) (Figure 1). The FHBi ranged from 0% to 64.0% in Radzików and from 0% to 54.0% in Poznań. The proportion of *Fusarium*-damaged kernels was, on average, FDKw = 6.9% and FDK # = 8.5%; this value was higher in Poznań (FDKw = 9.8%; FDK# = 12.0%) than in Radzików (FDKw = 5.0%; FDK# = 6.2%). The FDK ranged from 0% to 61.2% in Radzików and from 0% to 45.8% in Poznań for FDKw and from 0% to 62.1% in Radzików and from 0.1% to 61.5% in Poznań for FDK#. Differences between locations were statistically significant only for FDK# at $p = 0.013$.

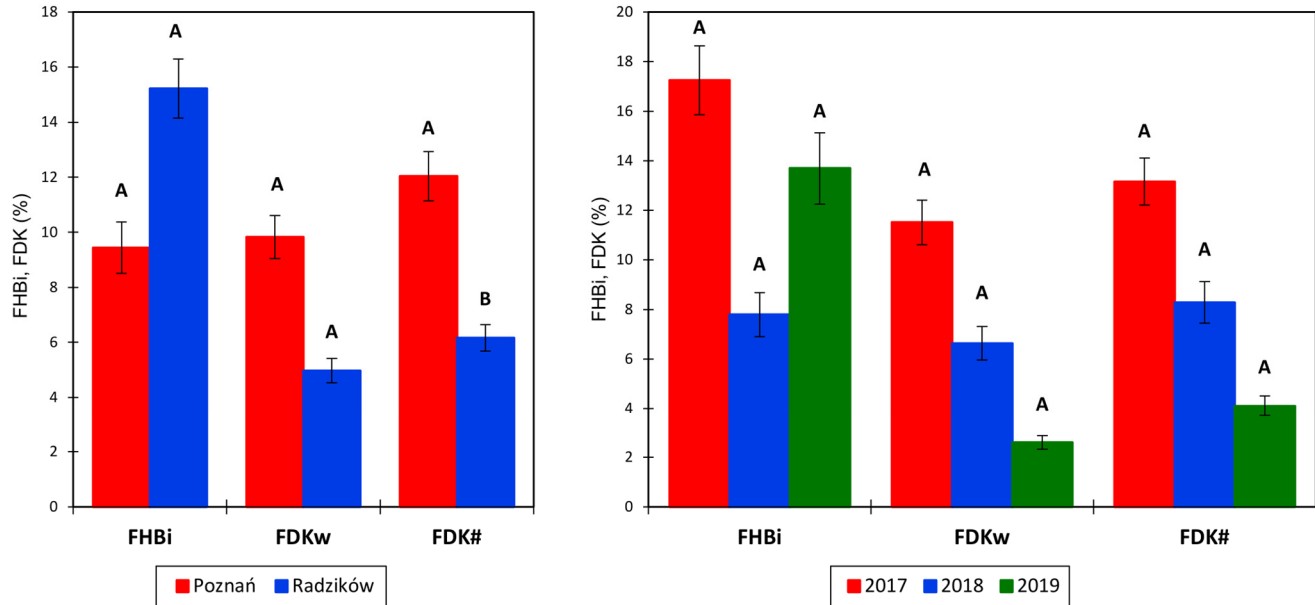

**Figure 1.** Average FHB index (FHBi) and *Fusarium*-damaged kernel percentage (FDK#—number, FDKw—weight) in two experimental locations (**left**) and three experimental years (**right**). The data are means ± SEM (*n* = 243 and *n* = 162). Means marked with the same letter are not significantly different at *α* = 0.05 according to Fisher's LSD test performed on the transformed variables (check ANOVA tables in Table 1).

**Table 1.** Analysis of variance of FHB index and *Fusarium*-damaged kernel percentage (in weight and number) for 27 wheat cultivars and lines in two experimental locations (Poznań, Radzików) and three years (2017, 2018, 2019).

| Source | DF | FHBi | | | FDKw | | | FDK# | | |
|---|---|---|---|---|---|---|---|---|---|---|
| | | Mean Squares | F | *p*-Value | Mean Squares | F | *p*-Value | Mean Squares | F | *p*-Value |
| Year | 2 | 42.854 | 1.776 | 0.364 | 33.213 | 9.005 | 0.101 | 28.157 | 10.282 | 0.089 |
| Location | 1 | 27.028 | 1.107 | 0.298 | 11.014 | 2.971 | 0.091 | 18.137 | 6.615 | 0.013 |
| Line | 26 | 13.845 | 17.868 | <0.0001 | 2.521 | 9.787 | <0.0001 | 3.234 | 11.324 | <0.0001 |
| Year × Location | 2 | 24.419 | 23.049 | <0.0001 | 3.707 | 13.442 | <0.0001 | 2.742 | 9.488 | 0.001 |
| Year × Line | 52 | 0.775 | 0.731 | 0.869 | 0.258 | 0.934 | 0.597 | 0.286 | 0.988 | 0.517 |
| Location × Line | 26 | 2.120 | 2.001 | 0.017 | 0.186 | 0.676 | 0.860 | 0.232 | 0.803 | 0.724 |
| Year × Location × Line | 52 | 1.059 | 3.867 | <0.0001 | 0.276 | 2.656 | <0.0001 | 0.289 | 2.166 | <0.0001 |
| Error | 234 | 0.274 | | | 0.104 | | | 0.133 | | |

During the three experimental years, the FHB index amounted to 17.2% in 2017, 7.8% in 2018, and 13.7% in 2019. The FDK proportion was 11.5% and 13.2% in 2017, 6.6% and 8.3% in 2018, and 2.6% and 4.1% in 2019 for FDKw and FDK#, respectively. Differences between years were not statistically significant.

The concentration of ERG in grain was, on average, 11.6 mg/kg and was higher in samples from Poznań; however, the concentrations did not differ significantly (Figure 2; Table 2). The concentration ranged from 0.5 to 49.5 mg/kg in Poznań and from 1.7 to 72.6 mg/kg in Radzików. The average ERG content in grain was the highest in 2017, lower in 2018, and the lowest in 2019. Differences between years were not statistically significant.

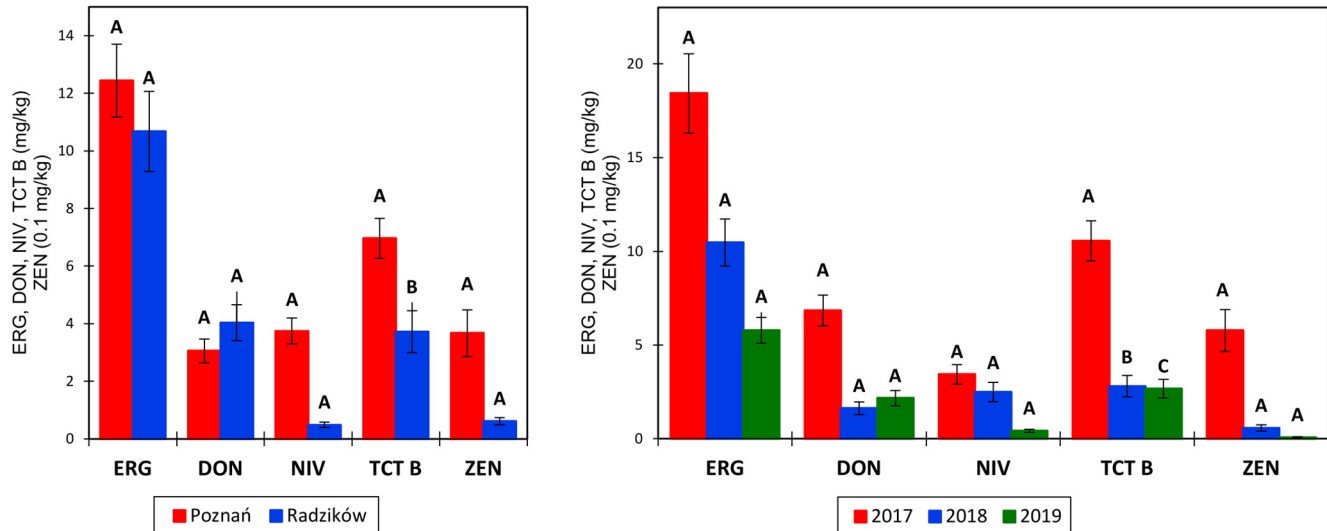

**Figure 2.** Average concentration of ergosterol (ERG), deoxynivalenol (DON), nivalenol (NIV), type B trichothecenes (TCT B), and zearalenone (ZEN) in grains of 27 wheat lines in two experimental locations (**left**) and three years (**right**). Values for 3AcDON and 15AcDON are not shown. The data are means ± SEM (*n* = 81 and *n* = 54). Means marked with the same letter are not significantly different at *α* = 0.05 according to Fisher's LSD test performed on the transformed variables (check ANOVA tables in Tables 2 and 3).

**Table 2.** Analysis of variance of concentrations of ergosterol, DON, 3ACDON, and 15AcDON in the grains of 27 wheat cultivars and lines in two experimental locations (Poznań, Radzików) and three years (2017, 2018, 2019).

| Source | DF | ERG | | | DON | | | 3AcDON | | | 15AcDON | | |
|---|---|---|---|---|---|---|---|---|---|---|---|---|---|
| | | MS | F | *p*-Value | MS | F | *p*-Value | MS | F | *p*-Value | MS | F | *p*-Value |
| Year | 2 | 2.544 | 3.919 | 0.206 | 3.829 | 3.929 | 0.208 | 0.113 | 16.451 | 0.071 | 0.007 | 58.386 | <0.0001 |
| Location | 1 | 0.076 | 0.115 | 0.766 | 0.239 | 0.241 | 0.672 | 0.007 | 0.957 | 0.431 | 0.000 | 1.000 | 0.423 |
| Line | 26 | 0.290 | 8.619 | <0.0001 | 0.247 | 6.954 | <0.0001 | 0.008 | 4.539 | <0.0001 | 0.000 | 1.000 | 0.485 |
| Y × Loc. | 2 | 0.655 | 16.641 | <0.0001 | 0.991 | 19.201 | <0.0001 | 0.007 | 3.377 | 0.042 | 0.000 | 0.069 | 0.934 |
| Y × Line | 52 | 0.034 | 0.855 | 0.713 | 0.036 | 0.688 | 0.909 | 0.002 | 0.824 | 0.757 | 0.000 | 0.624 | 0.954 |
| Loc. × Line | 26 | 0.034 | 0.857 | 0.658 | 0.018 | 0.344 | 0.998 | 0.001 | 0.397 | 0.994 | 0.000 | 1.000 | 0.485 |
| Error | 52 | 0.039 | | | 0.052 | | | 0.002 | | | 0.000 | | |

Y—year, Loc.—location.

**Table 3.** Analysis of variance for the concentrations of nivalenol, trichothecenes B (sum of DON, 3AcDON, 15AcDON, and NIV), and zearalenone (ZEN) in the grains of 27 wheat cultivars and lines in two experimental locations (Poznań, Radzików) and three years (2017, 2018, 2019).

| Source | DF | NIV | | | TCT B | | | ZEN | | |
|---|---|---|---|---|---|---|---|---|---|---|
| | | MS | F | *p*-Value | MS | F | *p*-Value | MS | F | *p*-Value |
| Year | 2 | 2.003 | 1.700 | 0.371 | 4.026 | 750.342 | <0.0001 | 47.632 | 3.113 | 0.243 |
| Location | 1 | 6.629 | 5.608 | 0.141 | 1.540 | 115.447 | 0.009 | 4.272 | 0.279 | 0.650 |
| Line | 26 | 0.136 | 8.328 | <0.0001 | 0.353 | 12.591 | <0.0001 | 1.297 | 3.815 | <0.0001 |
| Y × Loc. | 2 | 1.182 | 57.174 | <0.0001 | 0.013 | 0.285 | 0.753 | 15.318 | 43.171 | <0.0001 |
| Y × Line | 52 | 0.016 | 0.791 | 0.799 | 0.028 | 0.600 | 0.966 | 0.340 | 0.958 | 0.561 |
| Loc. × Line | 26 | 0.023 | 1.128 | 0.347 | 0.023 | 0.494 | 0.973 | 0.658 | 1.855 | 0.029 |
| Error | 52 | 0.021 | | | 0.047 | | | 0.355 | | |

Y—year, Loc.—location.

The amount of DON in grain was, on average, 3.543 mg/kg, ranging from 0 to 25.960 mg/kg, and the amount of NIV in grain was 2.115 mg/kg, ranging from 0 to17.400 mg/kg. Concentrations of DON in Radzików were higher than in Poznań; however, the difference was not statistically significant (Figure 2; Table 2). The concentration of NIV in Radzików was very low (0.485 mg/kg), about seven times lower than that in Poznań (3.756 mg/kg). However, differences between locations were not statistically significant. In the three experimental years, the amounts of DON and NV were as follows: 2017—6.846 and 3.436 mg/kg, 2018—1.622 and 2.491 mg/kg, and 2019—2.162 and 0.418 mg/kg, and years did not differ significantly (Table 2).

Acetylated derivatives of DON (3AcDON, 15AcDON) were detected at low amounts. On average, we detected 0.130 mg/kg of 3AcDON (0–1.375 mg/kg) and 0.016 mg/kg of 15AcDON (0–0.325 mg/kg). In Poznań, the concentrations of 3AcDON and 15AcDON were 0.145 and 0.017 mg/kg, respectively, and in Radzików, the concentrations were 0.114 and 0.015 mg/kg and did not differ significantly. In the three experimental years, the amounts of 3AcDON and 15AcDON were as follows: 2017—0.275 and 0 mg/kg, 2018—0.024 and 0.048 mg/kg, and 2019—0.090 and 0 mg/kg, and for 15AcDON, years differed significantly at $p < 0.001$.

The total amount of analyzed type B trichothecenes was 5.804 mg/kg, with a range of 0–30.854 mg/kg. In Poznań, the amount of TCT B was 6.962 mg/kg, with a range of 0.143–29.900 mg/kg, and in Radzików, the amount was 4.1646 mg/kg, with a range of 0–30.5845 mg/kg. Locations differed significantly at $p < 0.009$. In the three years, the average amount of TCT B was as follows: 2017—10.557 mg/kg, 2018—4.7185 mg/kg, and 2019—2.671 mg/kg and differed significantly at $p < 0.001$.

Zearalenone was detected in grain at an average amount of 0.214 mg/kg, ranging from 0 to 3.714 mg/kg. ZEN was present mainly in samples from Poznań in an amount of 0.367 mg/kg. In samples from Radzików, the ZEN concentration was five times lower and amounted to 0.061 mg/kg. Locations did not differ significantly. The highest concentration of ZEN was detected in 2017, at 0.578 mg/kg, followed by the concentration in 2018, at 0.057 mg/kg. In 2019, the concentration of ZEN was very low—0.008 mg/kg. Years differed significantly at $p < 0.001$.

Analysis of variance of the FHB index showed a very high effect of wheat line and no effect of year (random) and location (Table 1). No interactions of year × line were observed. However, highly significant interactions of year × location and year × location × line were found. Similarly, for FDKw and FDK#, the effect of the line was highly significant, as were the year × location and year × location × line interactions. Interactions of location × line were not significant for FDKs. For FDK#, the effect of location was significant at a low level.

The three-way interactions were significant; therefore, the effect of lines was analyzed separately for locations. For the results from Poznań, we found a significant interaction of year × line for the FHB index, but it was insignificant for FDKw and FDK#. Significance of interaction resulted from FHB indexes for susceptible lines DL 325/11/3, SMH 8694 and SMH 8816 and cultivar Patras. These three lines were the most infected in 2019, and cultivar Patras was also most infected in 2018. The other lines/cultivars were generally the most infected in 2017 and the least in 2019. For the results from Radzików, we found significant interactions of year × line for FHB index FDKw and FDK#. Wheat lines and cultivars had similar FHB indexes in 2017 and 2019, although in 2018, FHBi was significantly lower. Significance of interaction resulted from FHB indexes in 2018 for some medium-resistant lines/cultivars (POB 170/04, POB 679/03, RGT Kilimanjaro) which were more infected than susceptible cultivars Patras and Artist. *Fusarium*-damaged kernel proportions (FDKw, FDK#) were similar for most lines in 2017 and 2018 and lower in 2019. However, we found that the most resistant lines (S 10 [Fhb1+], S 30 [Fhb1+] and S 32 [Fhb1+]) had very low FDK in 2018. Two susceptible lines, SMH 8694 and SMH 8816, and cultivar Patras, had higher FDKs in 2019 than in 2018.

We observed no significant effect of year on the concentrations of all analyzed toxins, with exception of 15AcDON and the sum of trichothecenes (Tables 2 and 3). Location had a significant effect only on the concentrations of the sum of trichothecenes.

Location–mean square values for NIV were higher than the year–mean square values. This toxin was mainly detected in samples from Poznań (NIV) (Figure 2). The effect of the wheat line was highly significant for ERG and trichothecenes (except for 15AcDON).

We found significant interactions of year x location for ERG, DON, 3AcDON, NIV and ZEN. Interactions of year × line were non-significant for all metabolites and interactions of location × line were significant only for ZEN. Analysis of year x location interactions for ERG, DON and NIV showed that their significance resulted from unusual metabolite patterns in 2019. ERG concentration was higher in Radzików than in Poznań in 2019, but lower in 2017 and 2018. In contrast, DON concentrations were higher in Poznań than in Radzików in 2019 but lower in 2017 and 2018. Regarding NIV, the concentrations were about 5 and 60 times higher in Poznań than in Radzików in 2017 and 2018, respectively. In 2019, the amounts in Poznań and Radzików were similarly low; however, in Radzików, it was higher than in 2018.

Concentration of ZEN was very high in Poznań in 2017 and 14 times higher than in Radzików. In the next two years, concentration of ZEN in grain was low; however, in both years, it was about four times higher than in Poznań.

The winter wheat line 'KBP 14 16' showed the highest value for the FHB index. The lowest FHBi was observed in lines carrying the *Fhb1* resistance gene (UNG 136.6.1.1 [Fhb1+], S 10 [Fhb1+], and S 32 [Fhb1+]) and in lines without *Fhb1* resulting from crosses with resistant genotypes ('A40-19-1-2', '20828') (Table 4). Among breeding lines, the lowest FHBi was observed for four lines: SMH 7983, NAD 13014, NAD 13017, and STH 9059. Highly infected heads had three lines: DL325/11/3, SMH 8694, and SMH 8816.

**Table 4.** FHB index and *Fusarium*-damaged kernel percentage (in weight and number) for 27 winter wheat lines and cultivars in two experimental locations (Poznań, Radzików) and three years (2017, 2018, 2019).

| Line | FHBi (%) | FDKw (%) | FDK# (%) |
|---|---|---|---|
| KBP 14 16 | 41.6 a | 25.2 a | 28.7 a |
| DL325/11/3 | 37.0 ab | 17.5 ab | 20.4 ab |
| SMH 8694 | 35.5 ab | 10.3 bc | 12.9 bc |
| SMH 8816 | 32.8 ab | 8.9 cd | 11.0 cd |
| NAD 10079 | 28.7 abc | 9.1 bc | 11.2 bc |
| Patras | 22.1 bcd | 9.3 cd | 13.3 cd |
| Artist | 20.0 cde | 10.2 bc | 13.4 bc |
| RGT Kilimanjaro | 14.4 def | 5.0 defg | 7.3 def |
| Arina | 10.9 efgh | 6.8 cde | 8.0 cde |
| POB 679/03 | 10.1 fghi | 4.4 efgh | 4.0 ghijk |
| POB 0616 | 9.8 ghi | 6.9 defg | 8.4 efg |
| STH 008 | 9.7 fghi | 7.2 defg | 8.9 def |
| NAD 13024 | 8.9 efgh | 5.8 efg | 7.5 efg |
| AND 82/11/50 | 8.8 efg | 6.7 cdef | 9.3 cde |
| POB 170/04 | 8.6 ghi | 4.6 efg | 4.7 fghi |
| AND 4023/14 | 7.1 ghi | 4.7 efgh | 6.8 efg |
| SMH 7983 | 6.6 ghij | 5.8 efgh | 7.3 efgh |
| NAD 13014 | 5.6 ghij | 4.1 fghi | 5.3 efghi |
| Fregata | 5.1 ghij | 5.9 efg | 7.7 efg |
| NAD 13017 | 4.9 hij | 4.6 efgh | 5.9 efghi |
| STH 9059 | 4.7 ij | 3.9 ghij | 4.6 fghij |
| S 30 [Fhb1+] | 3.8 j | 3.4 jk | 4.2 kl |
| UNG 136.6.1.1 [Fhb1+] | 2.9 j | 3.6 ghijk | 4.2 hijkl |
| S 10 [Fhb1+] | 2.4 j | 3.6 hijk | 4.5 ijkl |
| S 32 [Fhb1+] | 2.3 j | 2.1 k | 2.4 l |
| A40-19-1-2 | 2.2 j | 4.2 ghijk | 5.0 ghijk |
| 20828 | 2.1 j | 2.6 ijk | 2.9 jkl |
| Means | 12.9 | 6.9 | 8.5 |

Means marked with the same letter are not significantly different at $\alpha = 0.05$ according to a Fisher LSD test performed on the Box–Cox-transformed variables; means ranked by FHBi values.

The lowest FDK proportions (in weight and number) were observed for low FHB-infected lines carrying the *Fhb1* gene, as well as for line 20828. Line A40-19-1-2 had the highest FDK values. Among the breeding lines, the lowest kernel damage was found for the five lines POB 679/03, STH 9059, A40-19-1-2, POB 170/04, and NAD 13014. FHB-susceptible lines (DL 325/11/3 and KBP 14 16) also presented high FDK values. The two high-yielding cultivars Artist and Patras also showed high levels of kernel damage. The third cultivar, RGT Kilimanjaro, despite having a similar head infection, exhibited FDK values two times lower than the previous two cultivars.

Ergosterol concentration was the highest in grains of the susceptible wheat lines KBP 14 16 and DL 325/11/3 (Table 5). Ergosterol concentration was also high in grains of the cultivars Artist and Patras that had high levels of kernel damage, as well as in grains of the 'RGT Kilimanjaro' cultivar that had low levels of kernel damage. The lowest ERG content was found in the grains of lines with the *Fhb1* gene, S 10 [Fhb1+] and S 30 [Fhb1+], as well as in the resistant lines A40-19-1-2 and 20828. For breeding lines, the lowest ERG concentration was found in grains of STH 9095 and SMH 7983.

**Table 5.** Concentrations of ergosterol (mg/kg), type B trichothecenes (DON, 3AcDON, 15AcDON, NIV, and TCT B) (mg/kg), and zearalenone (ZEN) (mg/kg) in the grains of 27 winter wheat cultivars and lines in two experimental locations (Poznań, Radzików) and three years (2017, 2018, 2019).

| Line | ERG | DON | 3Ac DON | 15Ac DON | NIV | TCT B [a] | ZEN |
|---|---|---|---|---|---|---|---|
| KBP 14 16 | 30.8 a | 9.344 ab | 0.453 a | 0.035 | 6.326 a | 16.158 a | 0.851 ab |
| DL325/11/3 | 22.7 ab | 11.698 a | 0.453 a | 0.021 | 3.426 abc | 15.598 a | 0.581 a |
| Artist | 22.8 abc | 6.780 b–e | 0.275 b | 0.026 | 4.629 ab | 11.710 abc | 0.485 b–e |
| Patras | 25.9 abc | 5.910 c–g | 0.232 bcd | 0.021 | 4.983 ab | 11.147 b–e | 0.245 cde |
| SMH 8694 | 20.0 ab | 6.181 abc | 0.232 bc | 0.030 | 3.876 abc | 10.318 ab | 0.421 a–d |
| SMH 8816 | 17.1 ab | 5.597 a–d | 0.189 b–e | 0.043 | 4.348 abc | 10.177 ab | 0.335 a–d |
| NAD 10079 | 16.3 a–e | 4.772 a–d | 0.122 b–g | 0.024 | 3.959 abc | 8.877 bcd | 0.267 abc |
| AND 82/11/50 | 11.3 c–h | 4.228 c–g | 0.153 b–f | 0.000 | 2.252 b–e | 6.633 d–g | 0.141 cde |
| RGT Kilimanjaro | 17.6 b–f | 3.275 c–h | 0.125 b–g | 0.015 | 2.816 a–d | 6.232 d–g | 0.349 cde |
| Fregata | 9.8 e–h | 4.472 c–f | 0.100 c–f | 0.010 | 1.145 def | 5.727 e–h | 0.062 de |
| Arina | 10.9 c–g | 3.783 c–f | 0.075 efg | 0.008 | 1.638 c–f | 5.503 c–f | 0.276 cde |
| NAD 13017 | 9.0 d–h | 2.858 c–h | 0.118 b–g | 0.000 | 1.359 def | 4.335 f–i | 0.152 de |
| NAD 13024 | 7.4 f–i | 3.013 c–h | 0.102 c–g | 0.000 | 1.195 def | 4.311 f–i | 0.124 cde |
| NAD 13014 | 7.3 f–i | 2.356 d–i | 0.087 d–g | 0.000 | 1.623 def | 4.065 f–j | 0.273 de |
| POB 0616 | 13.2 d–h | 2.556 f–j | 0.079 efg | 0.008 | 1.308 def | 3.951 f–k | 0.108 cde |
| UNG 136.6.1.1 [Fhb1+] | 6.9 ghi | 1.678 g–j | 0.048 fg | 0.008 | 1.960 def | 3.695 h–l | 0.108 de |
| AND 4023/14 | 5.7 ghi | 2.092 e–j | 0.092 d–g | 0.000 | 1.332 def | 3.516 f–k | 0.085 cde |
| STH 008 | 7.7 f–i | 2.478 c–h | 0.114 c–g | 0.008 | 0.861 ef | 3.461 f–j | 0.092 de |
| POB 170/04 | 6.4 ghi | 2.224 d–i | 0.092 d–g | 0.021 | 0.861 ef | 3.197 f–k | 0.111 cde |
| SMH 7983 | 5.3 ghi | 1.672 f–j | 0.076 efg | 0.000 | 1.136 def | 2.883 g–k | 0.193 de |
| POB 679/03 | 6.8 f–i | 2.065 e–j | 0.069 efg | 0.029 | 0.610 ef | 2.773 g–k | 0.073 cde |
| A40–19–1–2 | 5.1 i | 1.388 hij | 0.033 fg | 0.000 | 1.340 def | 2.762 k–l | 0.052 e |
| STH 9059 | 5.3 hi | 1.705 f–j | 0.073 efg | 0.004 | 0.734 ef | 2.516 i–l | 0.218 cde |
| 20828 | 5.5 i | 1.031 hij | 0.031 fg | 0.049 | 1.393 def | 2.504 jkl | 0.047 e |
| S 30 [Fhb1+] | 4.6 i | 1.238 hij | 0.034 fg | 0.054 | 0.620 ef | 1.946 kl | 0.059 de |
| S 10 [Fhb1+] | 4.5 i | 0.561 j | 0.017 g | 0.000 | 0.912 ef | 1.489 l | 0.047 e |
| S 32 [Fhb1+] | 6.2 hi | 0.714 ij | 0.024 fg | 0.021 | 0.468 f | 1.226 l | 0.028 e |
| Means | 11.6 | 3.543 | 0.129 | 0.016 | 2.115 | 5.804 | 0.214 |

[a] sum of DON, 3AcDON, 15AcDON, and NIV; means marked with the same letter are not significantly different at $\alpha = 0.05$ according to a Fisher LSD test performed on $\log_{10}$-transformed variables; means are ranked by TCT B concentration.

The largest amount of DON accumulated in the grains of susceptible wheat lines (DL 325/11/3 and KBP 14 16). A large amount of DON was also detected in the grains of two other susceptible lines (SMH8694 and SMH 8816) and the two check cultivars Artist and Patras. The amount of DON in the grains of RGT Kilimanjaro was two times lower. The lowest concentration of DON was detected in the grains of the five resistant check lines. DON levels were higher only in grains of the line UNG 136.6.1.1 [Fhb1+]. In grains

of the breeding lines, the DON amount was lowest for the two lines with the lowest ERG concentrations: STH 9095 and SMH 7983. Nivalenol was present mainly in grains of the susceptible lines KBP 14 16 and SMH 8816, as well as in grains of the cultivars Artist and Patras. Nivalenol content was lower in grains of the DL 325/11/3 line. Similar NIV amounts were detected in the grains of two cultivars (RGT Kilimanjaro and Arina) and the resistant lines UNG 136.6.1.1 [Fhb1+] and 20828. The lowest concentration of NIV was detected in the grains of three resistant wheat lines with the *Fhb1* gene and four breeding lines (STH 008, POB 170/04, STH 9059, and POB 679/03). Trichothecene 3AcDON was detected mainly in susceptible wheat lines and cultivars that accumulated large amounts of DON and NIV. The concentration of 15AcDON in grain was very low.

The total amount of the four type B trichothecenes was the highest in the grains of the four susceptible lines and the two cultivars Artist and Patras. Type B trichothecene levels were the lowest in the grains of the three lines with the *Fhb1* gene. Type B trichothecene levels were also low in the grains of the three breeding lines SMH 7983, POB 679/03, and STH 9059.

Differences in ZEN concentrations between wheat lines were of low significance (Table 6). The lowest amount of ZEN was found in the grains of resistant lines S 32 [Fhb1+], S 10 [Fhb1+], and 20828. A low amount of ZEN (below 0.100 mg/kg) accumulated in the grains of lines POB 0616, STH 008, AND 4023/14, and POB 679/03 and in the cultivar Fregata. The highest amount of ZEN (on average, 0.851 mg/kg; maximum, 3.714 mg/kg) was found in grains of the susceptible line KBP 14 16.

The FHB index was correlated significantly with other variables except for 15AcDON (Table 6). The coefficient of correlation of FHBi with DON had a higher value than the coefficient of correlation with NIV. *Fusarium*-damaged kernel proportions (in weight and number) correlated highly significantly with the concentration of mycotoxins. The lowest values had coefficients of correlation with NIV and the highest with DON and 3AcDON. The ergosterol concentration in grain correlated significantly with all mycotoxins except for 15AcDON. The lowest values had coefficients of correlation with ZEN. Trichothecene toxins correlated significantly with each other (except for 15AcDON) and with ZEN. The lowest value had coefficient of correlation of DON vs. NIV.

**Table 6.** Coefficients of correlation between the FHB index, *Fusarium*-damaged kernel proportion (in weight and number), and concentrations of ergosterol and mycotoxins in the grains of 27 winter wheat lines in two experimental locations (Poznań, Radzików) and three years (2017, 2018, 2019).

| Variables | FHBi | FDKw | FDK# | ERG | DON | 3Ac DON | 15Ac DON | NIV | TCT B |
|---|---|---|---|---|---|---|---|---|---|
| FDKw | 0.894 | | | | | | | | |
| FDK# | 0.884 | 0.984 | | | | | | | |
| ERG | 0.883 | 0.853 | 0.868 | | | | | | |
| DON | 0.900 | 0.919 | 0.924 | 0.911 | | | | | |
| 3AcDON | 0.837 | 0.922 | 0.908 | 0.854 | 0.911 | | | | |
| 15AcDON | $0.319^{ns}$ | $0.177^{ns}$ | $0.113^{ns}$ | $0.291^{ns}$ | $0.201^{ns}$ | $0.256^{ns}$ | | | |
| NIV | 0.802 | 0.797 | 0.827 | 0.902 | 0.826 | 0.796 | $0.291^{ns}$ | | |
| TCT B | 0.895 | 0.907 | 0.923 | 0.941 | 0.978 | 0.900 | $0.233^{ns}$ | 0.924 | |
| ZEN | 0.840 | 0.810 | 0.818 | 0.798 | 0.833 | 0.807 | $0.111^{ns}$ | 0.806 | 0.855 |

Coefficients are significant at $p < 0.001$ except for those marked with $^{ns}$ (non-significant).

Some lines showed higher toxin (TCT B) content than could be expected from FHB index values. These were lines UNG 136.6.1.1 [Fhb1+], NAD 13017, Fregata, NAD 13014, and 82/11/50. All these lines showed higher levels of FDK or ERG concentration compared with similarly infected lines. Some lines showing higher concentrations of trichothecene toxins (TCT B) than similarly infected lines had higher contents of NIV in grain; for example, lines 20828, A40-19-1-2, UNG 136.6.1.1 [Fhb1+], NAD 13017, NAD 13014, and 82/11/50. There were also lines with a low content of toxins (TCT B) despite their high FHB index. These were lines POB 679/03, POB 170/04, and STH 008. The first two lines had low FDK

values and ERG contents in grain. All three lines showed low concentrations of NIV in the grain.

Multivariate PCA analysis showed that the highest FHB resistance described with six traits (FHBi, FDK#, ERG, DON, NIV, and ZEN) was found in resistant wheat lines carrying the *Fhb1* gene and in two lines without this gene (20828 and A40-19-1-2) (Figure 3). Among the breeding lines, five other lines also showed considerable FHB resistance (POB 679/03, POB 170/04, and 4023/14, STH 9095, and SMH 7983). The most susceptible were lines DL 325/11/3 and KBP14 16, which accumulated high amounts of DON and ZEN and had significant kernel damage.

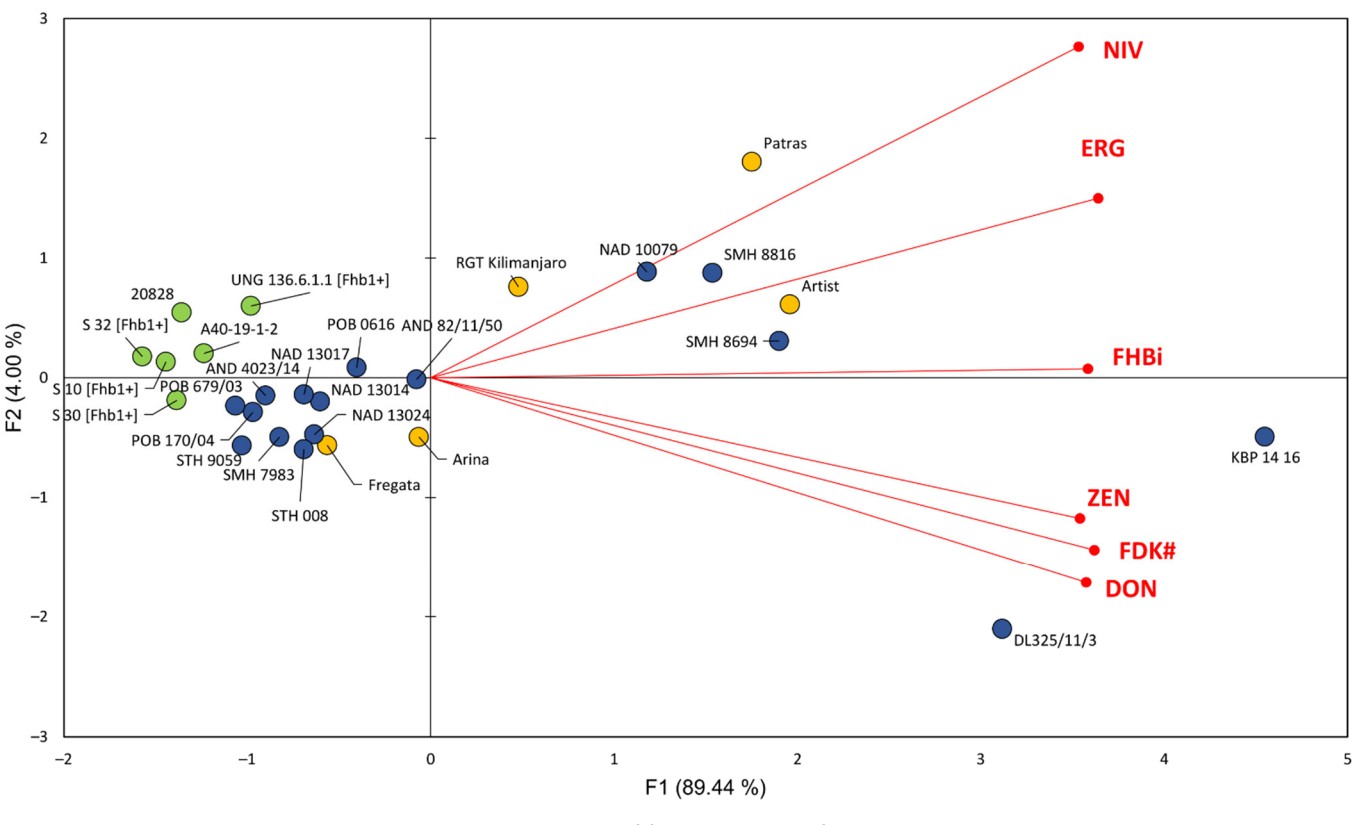

**Figure 3.** Biplot of the principal component analysis (PCA) of FHBi and FDK# and the concentrations of ERG, DON, NIV, and ZEN in grain of 27 winter wheat lines and cultivars in two experimental locations (Poznań, Radzików) and three years (2017, 2018, and 2019). FDK weight, 3AcDON, and 15AcDON are not shown. Highly resistant lines—green circles, breeding lines—blue circles, cultivars—orange circles.

## 4. Discussion

Research has shown that environmental conditions affect the development of FHB and the accumulation of toxins in the grain [13,68]. FHB severity, kernel damage, and the concentration of *Fusarium* metabolites were found to be affected by the experimental year and location. We applied mixed models to the ANOVA with year as a random factor. This resulted in insignificance of effects of year and location for the most of variables (except FDK# and TCT B) despite large differences between locations and years.

This study on resistance to FHB and the accumulation of *Fusarium* toxins was conducted over three years in two locations. To maintain humidity during and after inoculation, mist irrigation was used in Poznań. Although the infection of heads was higher in Radzików than in Poznań, the other parameters examined (the percentage of FDK and

amount of ERG and toxins; the sum of type B trichothecenes and zearalenone) were higher in Poznań. The only exception was the DON concentration, which was higher in Radzików.

In 2017, the weather in May was similar in both locations, and in June, rainfall in Radzików was two times greater than that in Poznań (Table S1). This rainfall led to a higher head infection rate in Radzików. In July, rainfall in Poznań was double that in Radzików. This caused high levels of kernel damage in Poznań—two times higher than that in Radzików. The amounts of type B trichothecenes in the grains in both locations were similar, although the amount of ZEN in grains in Poznań was very high (1.077 mg/kg) but very low in Radzików (0.079 mg/kg). We observed differences in the accumulation of DON and NIV in both locations. DON was mainly found in grain from Radzików (8.256 mg/kg versus 5.435 mg/kg in Poznań) and NIV was mainly found in samples from Poznań (5.786 mg/kg versus 1.087 mg/kg in Radzików).

The weather in 2018 was less favorable for FHB development than that in 2017, and rainfall in June was low compared to that in previous years. In both locations, head infection was low, despite the application of mist irrigation in Poznań. Kernel damage levels were lower than those in previous years and two times lower in Radzików than in Poznań. The same was found for the sum of type B trichothecenes, which was two times lower. As in 2017, we observed opposite results for DON and NIV concentrations, with a very low amount of DON observed in samples from Poznań and a very low amount of NIV in samples from Radzików. The ZEN concentration was 10 times lower than that in 2017 and 5 times higher in Radzików than in Poznań.

The weather conditions in Poznań in 2019 were unfavorable for the development of FHB. In Poznań, rainfall was very low in June, which was a period of wheat flowering and inoculations. The second half of June experienced very high temperatures, reaching 39 °C. The use of a mist irrigation system enabled inoculation of the heads, but symptoms were low. At the beginning of June, the precipitation in Radzików was much higher, which enabled effective inoculation and the appearance of symptoms of head infection (higher than those in Poznań). In the third decade of June, there was no rainfall, and air temperatures were very high (up to 38 °C). Weather conditions inhibited the development of FHB and the formation of mycotoxins, whose concentrations were the lowest over three years.

The observed differences in the accumulation of DON and NIV in the two locations were likely the result of competition between isolates of different chemotypes. Competition between *F. culmorum* and *F. graminearum* species was described by Van der Ohe and Miedaner [69]. One isolate of *F. graminearum* was of the NIV chemotype, and showed similar pathogenicity to isolates of the DON chemotype. Mixed DON+NIV chemotypes had stable pathogenicity (head infection) but varied in mycotoxin production under experimental conditions. The NIV chemotype is generally considered less aggressive than DON (3ADON, 15ADON) chemotypes [59,60]. The results of our experiments showed that this chemotype can produce considerable amounts of NIV, even in mixtures with more aggressive isolates of the 3ADON chemotype. However, this chemotype occurred mainly in Poznań (based on the production of NIV), where the application of mist irrigation created conditions more favorable for FHB development.

Wheat breeding lines showing low susceptibility to FHB were identified. The disease symptoms (on heads and kernels —-FHBi, FDK#, FDKw) in these lines were low and similar to the resistant checks (with or without the *Fhb1* gene). For example, these were the lines STH 9059, NAD 13017, and NAD 13014. However, for the trichothecene toxins in grains, the three lines with the *Fhb1* gene showed the lowest accumulation. The best breeding lines accumulated more trichothecenes. The amount of trichothecenes was similar in the two resistant checks without the *Fhb1* gene (20828 and A40-19-1-2). The first check was a progeny created by crossing the Capo winter wheat cultivar with Sumai 3. However, this progeny did not carry the resistance gene *Fhb1* [57].

We observed that some resistant lines (based on FHBi, FDK, ERG, DON) accumulated higher amounts of NIV in the grain. This resulted in the higher total accumulation of

trichothecenes type B. These were lines without *Fhb1* genes (20828, A40-19-1-2, and 4023/14) but also with the gene UNG 136.6.1.1 [Fhb1+]. It was found by Eudes et al. [70] that NIV is less phytotoxic than DON. This can explain low disease symptoms (head, kernels) in these lines despite high amounts of NIV in grain. This resulted in lower coefficients of correlation of FHBi and FDK with NIV than those found for DON. Miedaner et al. [71] found a low association between DON and NIV in wheat ($R^2 = 0.53$); however, in rye, it was very high ($R^2 = 0.92$). In our study, the determination coefficient was $R^2 = 0.65$. The significant deviation from the mean regression showed susceptible line DL 325/11/3 and medium resistant cultivar Fregata which accumulated four times more DON than NIV.

Line UNG 136.6.1.1 [Fhb1+] with the introduced *Fhb1* resistance gene showed low head infection, kernel damage and low accumulation of DON and 3AcDON. However, it accumulated more NIV than DON in grain. Other three lines with *Fhb1* gene had very low NIV accumulation. They had completely different genetic background which had an influence on the expression of the *Fhb1* gene. The other question is the effectiveness of the *Fhb1* gene in protection against NIV, which was not thoroughly studied. Only Lemmens et al. [72] found that *Fhb1* reduced FHB symptoms after application to wheat heads of DON as well as NIV. However, they postulated that the mechanism of resistance to NIV is different from that for DON. Thus, they could not confirm that *Fhb1* also provides resistance to NIV.

The most effective FHB resistance gene is *Fhb1* (*Qfhs.ndsu-3BS*), derived from the Sumai 3 cultivar. In various studies, this gene explained between 16% and 60% of the variability in the spread of pathogens in head tissue (type II resistance). This gene is commonly used in resistance breeding. The molecular marker UMN10, which is closely linked with the *Fhb1* gene, was previously developed to reliably identify the presence of the gene in breeding materials [41]. However, the number of commercial cultivars with the *Fhb1* gene remains limited. This gene is found mainly in varieties of spring wheat grown in the United States, Canada, and China [73,74] and is not found in European winter wheat cultivars. The only winter cultivar with *Fhb1*, Jaceo, was withdrawn from the market [75]. Currently, there is one cultivar of winter wheat with *Fhb1*, MS INTA 416, which is grown in Argentina [76].

The lack of success in introducing the *Fhb1* gene to intensive winter European cultivars may be due to the fact that, despite the use of marker-assisted selection (MAS), the presence of this gene has a negative impact on yield and quality characteristics [77,78]. Another factor hindering the introduction of the *Fhb1* gene (and others such as *Fhb2* and *Fhb5*) is the significant influence of the genetic background of the recipient genotype on the expression of this gene [50]. Genetic studies of the European winter wheat population have shown a lack of QTLs, and a strong effect is associated with FHB resistance [79–82]. For example, no QTL was found in the *Fhb1* region, which confirms the lack of variation in FHB resistance at the *Fhb1* locus. However, in the above studies, numerous low-effect QTLs have been identified on all wheat chromosomes. The presence of only low-effect QTLs hinders an effective strategy for pyramiding FHB resistance.

In addition, the morphological characteristics had a large impact on the degree of wheat infection by FHB and the results of QTL mapping studies. These characteristics were plant height, length of the peduncle, awn length, head compactness, and anther extrusion or retention. A strong link between the height of the wheat plants and the severity of FHB was emphasized [83–85], mainly due to differences in the microclimate at the level of heads in low and tall wheat canopies. However, a genetic link was found between FHB susceptibility and the presence of the dwarf genes *Rht1* (*Rht-B1*) and *Rht2* (*Rht-D1*) [86,87]. The mechanism underlying this phenomenon, however, is not fully known. It could be explained by the effects that these genes have on type I resistance by reducing anther extrusion. Retention of the anthers in flowers increases susceptibility to infection and stimulates mycelial growth, while selecting wheat lines towards open flowering can increase resistance to primary infection. The recently described dwarf gene *Rht24*, which does not affect the type of wheat flowering, may also be used [88,89].

## 5. Conclusions

In this study, we observed wide variability in the reactions of winter wheat lines to FHB. Visual observations of head infection and kernel damage were found to be reliable indicators of *Fusarium* toxin accumulation and can be applied for selection in resistance breeding. However, more reliable identification of resistant genotypes requires the additional analysis of *Fusarium* metabolites (at least DON).

Lines combining all types of FHB resistance were identified. However, lines with the highest FHB resistance accumulated more toxins than highly resistant check lines with the *Fhb1* gene introduced.

**Supplementary Materials:** The following are available online at https://www.mdpi.com/article/10.3390/agronomy11091690/s1, Table S1: Air temperature and rainfall in May, June, and July of 2017, 2018, and 2019 in the two experimental locations.

**Author Contributions:** Conceptualization, T.G., P.O., and H.W.; methodology, H.W., T.G. and P.O.; validation, T.G.; formal analysis, T.G.; investigation, H.W., T.G., P.O. and A.T.; resources, H.W., T.G. and P.O.; data curation, H.W., T.G. and P.O.; writing—original draft preparation, P.O., H.W. and T.G.; writing—review and editing, H.W., T.G., P.O. and AT.; visualization, T.G.; project administration, T.G.; funding acquisition, T.G. All authors have read and agreed to the published version of the manuscript.

**Funding:** Research was supported by the project from the Ministry of Agriculture and Rural Development: "Identification, and application of phenotypic, metabolic and molecular markers in studies of types of resistance to Fusarium head blight in winter wheat accessions differing in resistance", project no. 6 (HOR hn 801-PB-13/16, HOR.hn.802.28.2017, HOR.hn.802.19.20l8).

**Institutional Review Board Statement:** Not applicable.

**Informed Consent Statement:** Not applicable.

**Data Availability Statement:** Data supporting the reported results can be found at https://data.mendeley.com/datasets/sp36ghdx9k/1 (accessed on 24 August 2021).

**Acknowledgments:** The Authors wish to thank the wheat breeders from DANKO Hodowla Roślin Ltd., Hodowla Roślin Smolice Ltd.—IHAR-PIB Group, Hodowla Roślin Strzelce Ltd.—IHAR-PIB Group, Małopolska Hodowla Roślin Ltd., and Poznańska Hodowla Roślin Ltd. for providing winter wheat lines for the experiment.

**Conflicts of Interest:** The authors declare no conflict of interest.

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
