# Peer review of "Resistance to Fusarium Head Blight, Kernel Damage, and Concentrations of Fusarium Mycotoxins in the Grain of Winter Wheat Lines"

_agronomy, doi:10.3390/agronomy11091690_

Round 1
Reviewer 1 Report
The manuscript gives a good contribution of Resistance to Fusarium head blight, kernel damage, and concentrations of Fusarium mycotoxins in wheat. However, I have concerns about the methods and results of the paper that I believe need to be addressed in order to improve its clarity. In particular, statistical data of chromatographic analysis are missing. Their approach is interesting but it has some flaws that make this version unacceptable for publication. Provided they conduct changes to the manuscript, I believe this paper could be of interest to the interested reader on plant breeding.
A few points:
L.13: Delete “resistant”
L.15: Delete “The”
L.27: Keywords should be in alphabetic order. Also, keywords serve to widen the opportunity to be retrieved from a database. To put words that already are into title and abstracts makes KW not useful. Please choose terms that are neither in the title nor in abstract.
L.29: …to FHB and accumulation….
L.34: …and zearalenone are…
L.41: Sentence stating “In wheat, Fusarium head 41 blight (FHB) has…
L.43: …companies in the USA…
L.44: …the second in the USA and Canada…
L.46: Fusarium head blight should be abbreviated in FHB. Check in all manuscript.
L.47: Reference [9] not found. Revise.
Ls.47-48: Revise this sentence to eliminate wordiness.
Ls.53-54: Sentence is irrelevant. Delete of combine with anterior sentence.
L.65: Sentence starting “The FHB severity is characterized by different resistance…”
L.70-71: Rephrase by “Fusarium species produce numerous toxins that cause acute or chronic toxic effect in humans and animals…”
L.72: Again, “In wheat kernel, previous toxicological analysis revealed the presence of two…”
L.74: …zearalenone (ZEN)…
L:75: DON and NIV have…
L.77: … the World Health Organization (WHO),…
L.80: zearalenone should be abbreviated. Check in all manuscript.
L:83-85: Confusing, rephrase.
Ls.89-92: Revise this sentence to eliminate wordiness.
Ls.109-110: … can influence Fusarium species dissemination…
L.149: … under a 16:8 h (Light:Dark) photoperiod for…
Ls.205-216: More information about chromatographic analysis is needed. Explain.
L.221: Analysis of variance? But in results, I presume that data was analyzed by one, two, and three-way ANOVA. Please, specify what fixed effects are in each model. Also, what’s treatment mean comparison test was used?
L.237: In results, chromatographic analysis data are missing. Include.
Ls.238-265: Provide statistical values (F =?, DF =?, and p-values) for each ANOVA.
L.254: Delete “Conversely”
Figure 1 and 2 are confusing. The authors show two types of graphics: the first (left) has two treatments and must be analyzed by Student's test while the second (right) must be analyzed by ANOVA. However, this information must be analyzed by two-analysis ANOVA. Revise.
L.354: Deoxynivalenol should be abbreviated. Check in all manuscript.
Reviewer 2 Report
The manuscript describes extensively the results obtained in a three-years trial aimed to evaluate the FHB resistance in a panel of wheat accessions. The manuscript is highly descriptive, however the number of wheat accessions considered is limited and several are breeding lines. I suggest to reduce the manuscript lenght and all the details, that are of limited, local ineterst. In my opinion, the data obtained that can be of wider interest can be reported in a short communication form.
Round 2
Reviewer 1 Report
The authors have incorporated all suggestions and comments into the revised version, now the manuscript seems much clear. There are some minor points to be corrected:
Ls.42-43: In wheat, Fusarium Head Blight (FHB)…
Ls.241-242: Change “min.” by “min”
Ls. 496 and 499: Delete “significantly”
Ls.541-542: … NIV chemotype and showed…
L.590: …and stimulates mycelial growth, while selecting…
Author Response
Thank you for your comments. We have made suggested corrections and improved conclusions.
Reviewer 2 Report
Many thanks for your replay. I understand your point of view.
Author Response
Thank you for your positive report and comment to my replay.
This manuscript is a resubmission of an earlier submission. The following is a list of the peer review reports and author responses from that submission.
Round 1
Reviewer 1 Report
L 169 Why the Authors decided to spray inoculate the heads instead of using point inoculation? This is just my curiosity because you evaluated FHB parameters closer to the Type II resistance and it is well known that point inoculation is better toevaluate Type II resistance while spray inoculation is mosstly used for Type I resistance.
L 173 DOn't you think that evalueting the symptoms 3 weeks after the inoculation is too late and probably you lost some interesting information regarding the different responses of the wheat cultivars to FHB in the first 3 weeks?
L 187 Why did you use ELISA to quantify ZEN and gas chromatography for the other mycotoxins? Don't you think that ELISA could be less sensitive?
In all the graphs you should add the bars of the standard deviation or standard error
Reviewer 2 Report
Resistance to Fusarium head blight, kernel damage, and concentrations of Fusarium mycotoxins in the grain of winter wheat lines Ochodzki et al.
Comments to authors
The authors have evaluated winter wheat germplasm including breeding lines for Fusarium head blight and toxins in the grain. They found significant differences among the breeding lines for fusarium head blight and mycotoxins concertation. There was significant correlations between the FHB index and mycotoxins content. They have also found few breeding lines, which had similar level of resistance to the cultivars that contains fhb1 gene. The newly identified breeding lines can be used to develop winter wheat breeding cultivars with FHB resistance. This work seems important and manuscript is written fairly well. however, the it needs some improvements in writing especially in the result and need to look at the interaction further. The authors have mentioned there was three way (year x location x site) interaction significant for many variables. When there is three-way interaction significant, they might need to analyze the effect of lines for individual site-years separately to determine if the resistant and susceptible lines perform the same way in all site-years. And also, when there is a significant interaction the main effect does not make much sense unless they further investigate the interaction. There might be qualitative, crossover type, or quantitative interaction, difference in magnitude. if the interaction is qualitative the main effect is not very informative and it is not good to make any conclusion based on main effect.
The result section needs to improve as well. Instead of rewriting the numbers from the graph or table it would be better if they explained the mathematic relationships. Other comments are in the attached document.
Table titles – add location and year of the study in the table titles.
ANOVA tables – Provide actual P-values instead of asterisks.
